# Regulation of Immune Functions by Non-Neuronal Acetylcholine (ACh) via Muscarinic and Nicotinic ACh Receptors

**DOI:** 10.3390/ijms22136818

**Published:** 2021-06-24

**Authors:** Masato Mashimo, Yasuhiro Moriwaki, Hidemi Misawa, Koichiro Kawashima, Takeshi Fujii

**Affiliations:** 1Department of Pharmacology, Faculty of Pharmaceutical Sciences, Doshisha Women’s College of Liberal Arts, 97-1 Minamihokodate, Kodo, Kyotanabe, Kyoto 610-0395, Japan; mmashimo@dwc.doshisha.ac.jp; 2Department of Pharmacology, Keio University Faculty of Pharmacy, Minato-ku, Tokyo 105-8512, Japan; moriwaki-ys@pha.keio.ac.jp (Y.M.); misawa-hd@pha.keio.ac.jp (H.M.); 3Department of Molecular Pharmacology, Kitasato University School of Pharmaceutical Sciences, Minato-ku, Tokyo 108-8641, Japan; koichiro-jk@piano.ocn.ne.jp

**Keywords:** acetylcholine, muscarinic acetylcholine receptor, nicotinic acetylcholine receptor, lymphocyte, macrophage, secreted lymphocyte antigen-6/urokinase-type plasminogen activator receptor-related peptide-1, hippocampal cholinergic neurostimulating peptide, choline acetyltransferase

## Abstract

Acetylcholine (ACh) is the classical neurotransmitter in the cholinergic nervous system. However, ACh is now known to regulate various immune cell functions. In fact, T cells, B cells, and macrophages all express components of the cholinergic system, including ACh, muscarinic, and nicotinic ACh receptors (mAChRs and nAChRs), choline acetyltransferase, acetylcholinesterase, and choline transporters. In this review, we will discuss the actions of ACh in the immune system. We will first briefly describe the mechanisms by which ACh is stored in and released from immune cells. We will then address Ca^2+^ signaling pathways activated via mAChRs and nAChRs on T cells and B cells, highlighting the importance of ACh for the function of T cells, B cells, and macrophages, as well as its impact on innate and acquired (cellular and humoral) immunity. Lastly, we will discuss the effects of two peptide ligands, secreted lymphocyte antigen-6/urokinase-type plasminogen activator receptor-related peptide-1 (SLURP-1) and hippocampal cholinergic neurostimulating peptide (HCNP), on cholinergic activity in T cells. Overall, we stress the fact that ACh does not function only as a neurotransmitter; it impacts immunity by exerting diverse effects on immune cells via mAChRs and nAChRs.

## 1. Introduction

Although acetylcholine (ACh) has long been known as the classical neurotransmitter in the central and peripheral cholinergic nervous systems, molecular biological investigations have revealed its functions in a number of non-neuronal cholinergic systems. It is now known that non-neuronal cholinergic systems present in various tissues and organs are involved in diverse physiological and pathophysiological processes, including immune and inflammatory responses, wound healing, cancer development and progression, and cardiovascular, respiratory, digestive, and orthopedic diseases (for further details, please see the reviews [1,2,3]). In 2019, the 5th international symposium on non-neuronal ACh was held in Long Beach, CA, USA (27–29 September); presented were results from numerous ongoing studies on the non-neuronal cholinergic system in the context of diverse cells and organs (please see the proceedings “The 5th International Symposium on Non-neuronal Acetylcholine” [4]). Given the recent publication of extensive and comprehensive reviews on the “cholinergic anti-inflammatory pathway” [2,3,5], in this review we will focus mainly on the cholinergic system in immune cells.

### 1.1. ACh Synthesis by Choline Acetyltransferase in Immune Cells

ACh is synthesized from choline and acetyl coenzyme A in a reaction catalyzed by choline acetyltransferase (ChAT). The expression of ChAT was first identified in the nervous system [6,7]. However, the presence of ChAT has also been detected in a variety of non-neuronal cells and organs, including immune cells, keratinocytes, epithelial cells in the digestive and respiratory tracts, and the placenta [8]. The synthesis of ACh by ChAT in immune cells was first reported more than a quarter-century ago after reverse transcription-polymerase chain reaction (RT-PCR) and Western blot analysis were used to detect ChAT mRNA and protein expression in the MOLT-3 human leukemic T cell line (see 2.2. Cholinergic Components in Immune Cells), as will be discussed in more detail below.

### 1.2. Storage and Release of ACh in Immune Cells

Following ChAT-catalyzed synthesis in the cholinergic nervous system, ACh is loaded into synaptic vesicles via vesicular ACh transporters (VAChT) and is released from nerve endings by exocytosis triggered by elevation of the intracellular free Ca^2+^ concentration ([Ca^2+^]_i_) induced by nerve impulses. In lymphocytes, however, no structures resembling synaptic vesicles have been observed histologically. Moreover, Fujii et al. failed to detect gene expression of VAChT in human mononuclear leukocytes (MNLs), including T cells [9,10], which suggests that, within immune cells, ACh is not stored in structures akin to synaptic vesicles. Still, the possibility that ACh in T cells is localized within a storage apparatus of some kind cannot be ruled out. Consistent with that idea, nicotine was shown to cause an increase in plasma ACh and a decrease in the ACh content of blood cells in rabbits, which suggests the release of ACh from blood cells via one or more depolarization-independent pathways [11].

Mediatophore, an oligomer (16-kDa subunits) homologous to the proteolipid subunit c of vacuolar H^+^-ATPase (V-ATPase) located on the nerve terminal membranes of the *Torpedo* electric organ, is able to mediate Ca^2+^-dependent ACh translocation [12]. Interestingly, immunohistochemical analyses revealed the presence of a similar mediatophore in the cytoplasm and on the plasma membrane of two human T cell lines, CCRF-CEM and MOLT-3 cells [10]. Moreover, T cell activation induced by phytohemagglutinin (PHA) via T-cell receptors (TCRs) was associated with enhanced mRNA expression of the abovementioned mediatophore as well as with release of ACh. Although the precise mechanism by which the mediatophore regulates ACh release from T cells remains unclear [10], these observations are consistent with its involvement in ACh release from T cells.

### 1.3. ACh Receptors and Other Cholinergic Components

Within the nervous system, ACh acts on muscarinic and nicotinic ACh receptors (mAChR and nAChR, respectively) activate multiple intracellular signaling pathways to regulate diverse cellular functions. The action of ACh is terminated through its hydrolysis into choline and acetate catalyzed by acetylcholinesterase (AChE) or butyrylcholinesterase (BuChE) [13]. The choline is then taken up by the neuron via the high-affinity choline transporter (CHT1) as a source for subsequent ACh synthesis [14]. It is now evident that all components necessary for a cholinergic system, including ACh, ChAT, mAChRs, nAChRs, AChE, BuChE, and CHT1, are present in most immune cells and that the lymphocytic cholinergic system contributes to the regulation of various immune functions via mAChRs and nAChRs [2,3,15].

In the sections below, we will focus on (1) the expression of non-neuronal ACh in immune cells, (2) the signaling pathways activated via mAChRs and nAChRs in T cells, (3) the role of ACh in the regulation of antibody class switch, (4) the role of ACh in the regulation of macrophage function, and (5) the effects of two peptide cholinergic ligands on the cholinergic activity in immune cells.

## 2. Non-Neuronal ACh in the Immune Cells

### 2.1. ACh-Mediated Interaction of Vascular Endothelial Cells (VECs) with T Cells

Stimulation by ACh of M_3_ mAChRs (see Section 3.1) expressed on VECs activates nitric oxide (NO) synthesis, which in turn induces relaxation of vascular smooth muscles [16,17]. Furthermore, ChAT immunoreactivity was detected in VECs within the rat brain [18], suggesting ACh is synthesized by ChAT in VECs. In fact, both the synthesis and release of ACh have been demonstrated in bovine aortic endothelial cells and porcine cerebral microvessels, which suggests that ACh released from the VECs acts on mAChRs on the same cells’ surface in an autocrine fashion, leading to the production of NO [19,20] (Figure 1). In addition, the amount of ACh released by cultured VECs into the conditioned medium is significantly greater in the presence of isoflurophate (DFP), a non-competitive inhibitor of AChE and BuChE, than in its absence, though the ACh contents of VECs are comparable under the two conditions. This indicates that ACh synthesized in VECs is rapidly released and then extensively degraded by AChE and BuChE (Figure 1). That finding and the detection of ACh in the plasma and blood of various animal species, including humans, along with the development of highly sensitive radioimmunoassays [21], led to investigations into the origin of ACh in the blood and its physiological function (see also the reviews [15,22,23]).

VECs constitutively express major histocompatibility complex (MHC) I and II and a range of costimulatory molecules, including intercellular adhesion molecule-1 (ICAM-1) and vascular cell adhesion molecule-1 (VCAM-1), under the regulation of inflammatory cues (see a review by Carman and Martinelli [24]). VECs thus play an essential locally tuned role in the regulation of T cell migration and information exchange. For example, the interaction between VECs and CD4^+^ T cells via peptide-bound MHC (pMHC) II complexed with TCR induces transendothelial migration of T cells to sites of inflammation [25,26,27]. The interaction via adhesion molecules (VCAM-1 and ICAM-1) on VECs and very late antigen-4 (VLA-4) and lymphocyte function-associated antigen-1 (LFA-1) on T cells further facilitate T cell migration [25]. While the regulatory mechanisms governing ACh synthesis in VECs remain to be clarified, the synthesis and release of ACh from activated CD4^+^ T cells mediated via TCRs and adhesion molecules should be enhanced by antigen presentation [28]. We, therefore, suggest that ACh released from VECs and CD4^+^ T cells during their interaction acts at mAChRs and nAChRs on the cells in an autocrine or paracrine fashion, leading to NO synthesis by VECs and facilitation of T cell migration (Figure 2).

### 2.2. Cholinergic Components in Immune Cells

Studies of the expression and function of mAChRs and nAChRs in lymphocytes started in the early 1970s (see the reviews [15,22,23]). Stimulation of these receptors by their respective agonists was shown to elicit a variety of functional and biochemical effects, including enhancement of cytotoxicity and cell proliferation as well as the formation of cyclic GMP (cGMP) and IP_3_ (see the above reviews). Based on these findings, it was postulated at the time that the parasympathetic cholinergic nervous system might play a role in immune-neurohumoral crosstalk. However, ACh is far more degradable (enzymatically by AChE or BuChE and physicochemically in neutral and alkaline media) than other neurotransmitters such as catecholamines or serotonin [2,3,15,22,23].

The detection of signatures of cholinergic innervation of blood vessels as well as the parenchyma of the thymus [30,31] and bone marrow [32,33] suggested cholinergic control of lymphoid tissues. However, those findings were inconclusive, and large gaps in our understanding remain [34,35]. Its rapid enzymatic destruction dictates that ACh released from cholinergic nerve terminals can only work over a range of tens of nanometers [35]. We, therefore, hypothesize that activation of mAChRs and nAChRs on lymphoid tissues would require the formation of submicron synapses with cholinergic nerve terminals [34,35]. At present, no available histological evidence supports the formation of synapses between cholinergic nerve terminals and immune cells. It is, therefore, unlikely that ACh released from cholinergic nerve terminals diffuses to immune cells intact and acts on mAChRs and nAChRs (see the reviews [2,3,15,22,23]). For that reason, our focus is mainly on the effects of ACh derived from the immune cells themselves during cell-cell interactions via TCR-pMHC I or II and adhesion molecules [36].

The discovery of ChAT mRNA expression and the presence of ChAT protein in the MOLT-3 human leukemic T cell line by Fujii et al. [37] provided the first definitive evidence of ChAT-catalyzed ACh production in T lymphocytes. Subsequently, the constitutive expression of both ChAT mRNA and ACh was detected in human MNLs [9] and various human leukemic cell lines [38,39], and ChAT mRNA expression was confirmed in ACh-containing rat CD4^+^ and CD8^+^ T cells [40] and in mouse MNLs, dendritic cells (DCs) and macrophages [41]. Among human immune cells, T cells showed the highest ACh content and ChAT gene expression [39]. ChAT gene expression and ChAT-catalyzed ACh synthesis in immune cells, including CD4^+^ T cells, were further substantiated by detection of fluorescent reporter proteins in immune cells from ChAT BAC-eGFP transgenic mice [42] and ChAT-Cre-tdTomato mice [43]. Interestingly, the induction of ChAT mRNA expression in human MNLs activated by PHA suggests that antigen presentation via TCRs is involved in the regulation of T cell cholinergic activity through calcineurin-mediated pathways [9,28]. In addition, activation of protein kinase C (PKC) and protein kinase A (PKA) by phorbol 12-myristate 13-acetate (PMA) and dibutyryl cAMP, respectively, up-regulates ACh synthesis in T cells [28]. ACh released from the interacting immune cells would be expected to act on mAChRs and nAChRs in a paracrine and/or autocrine fashion to regulate cell functions, for instance during antigen presentation. In that regard, expression of mAChRs and nAChRs on immune cells, such as T cells, B cells, macrophages, and DCs, has been reported by several groups, including our laboratory [15,41]. In line with the above findings, recent publication by Hoover et al. [44] found the enhancement of ACh release from spleen cells by activation with isoproterenol, a β-adrenoceptor agonist and by TCR-activation with CD3/CD28 antibodies.

## 3. Ca^2+^ Signaling via mAChRs and nAChRs in T and B Cells

TCR stimulation by PHA elicits an elevation in [Ca^2+^]_i_ within T cells, which leads to activation of transcription regulators, including c-fos, and to modulation of lymphocyte function [45]. Stimulation of B-cell receptors (BCRs) similarly elicits [Ca^2+^]_i_ mobilization within B cells [46]. We previously showed for the first time that stimulation of mAChRs and nAChRs by their respective agonists induces Ca^2+^ signaling in both T and B cells [47,48,49], and this Ca^2+^ mobilization is a key signal involved in the regulation of various immune functions.

### 3.1. mAChR

Genes encoding five mAChR subtypes (M_1_–M_5_) have been identified [50,51]; the M_1_, M_3_, and M_5_ subtypes are coupled to phosphoinositide signaling pathways, whereas the M_2_ and M_4_ subtypes are linked to the adenylate cyclase system [52]. Expression of all five mAChRs subtypes has been detected in human MNLs and leukemic cell lines as well as mouse MNLs, DCs, and macrophages [41,53].

Ligand binding to transmitter receptors coupled to phosphoinositide signaling pathways activates phospholipase C and induces rapid, transient increases in [Ca^2+^]_i_ via IP_3_-induced release of Ca^2+^ from intracellular stores in the endoplasmic reticulum (ER) [54]. In addition, T cells express Ca^2+^ release-activated Ca^2+^ channels (CRACs), which mediate extracellular Ca^2+^ influx and regulate diverse immune functions [46]. Changes in [Ca^2+^]_i_, which are known to serve as an excitatory and/or inhibitory signal within cells [55], can be easily investigated using Ca^2+^-sensitive fluorescent probes (e.g., Fura-2) [56]. For instance, Kaneda et al. used selective agonists to show that stimulation of M_3_ mAChRs on populations of Fura-2-loaded Jurkat cells (T cells) causes transient elevations in [Ca^2+^]_i_ [57]. The oscillations evoked in a variety of cell types by prolonged exposure to an agonist are a clear example [54,55]. Oxotremorine-M (Oxo-M), an mAChR agonist, induces an initial transient increase in [Ca^2+^]_i_ followed by repetitive [Ca^2+^]_i_ oscillations in CCRF-CEM human leukemic T cells expressing M_3_ and M_5_ mAChRs, but not the M_1_ subtype ([58]; Figure 3). On the other hand, removal of the extracellular Ca^2+^ or pharmacological CRAC blockade abolishes the [Ca^2+^]_i_ oscillations without affecting the initial transient increase in [Ca^2+^]_i_ induced by Oxo-M (Figure 3). Pharmacological blockade of CRACs also suppresses Oxo-M-induced c-fos gene expression [58]. These findings suggest that upon M_3_ and/or M_5_ mAChR activation, IP_3_-induced Ca^2+^ release elicits an extracellular Ca^2+^ influx through CRACs, which generates repetitive [Ca^2+^]_i_ oscillations and, in turn, enhances c-fos gene expression in T cells.

Qian et al. demonstrated the plasticity of T cell AChRs; that is, the pattern and intensity of the mRNA expression of both mAChR and nAChR subtypes are altered by T cell activation via TCR-related pathways [59]. This suggests the pattern and intensity of mRNA expression of mAChR and nAChR subtypes may vary among individuals, depending on their immunological status, as was reported by Sato et al. [53].

### 3.2. nAChR

At present, 10 α (α 1–10), four β (β 1–4), and the γ, δ and ε nAChR subunits have been identified through molecular cloning. In skeletal muscle, the so-called muscle-type nAChRs contain four distinct subunits within a pentameric complex: (α1)_2_β1γδ or (α1)_2_β1εδ. On the other hand, both neuronal and non-neuronal cells express so-called neuronal-type nAChRs composed of only α subunits and βsubunits: eight α (α2–α7, α9, and α10) and three β(β2–β4) subunits. Immune cells express mainly the α2, α5, α6, α7, α9, α10, and β2 nAChR subunits [3,41,53]. The α8 subunit has been detected only in the visual areas of the avian brain [60]. Within the channel complex, at least two copies of the αsubunit are always present, and multiple binding sites for ACh are formed at the interface of an α and its neighboring subunit. nAChRs are ligand-gated ion channels permeable to Ca^2+^ as well as to Na^+^ and K^+^; their activation by agonists causes a rapid increase in the membrane permeability to Na^+^ and Ca^2+^, resulting in depolarization and cell excitation [46].

In CCRF-CEM cells, a human T cell line expressing the α3, α5, α6, α7, and β4 nAChR subunits [53], stimulation of nAChRs with nicotine induces an increase in [Ca^2+^]_i_ [49]. Spontaneous [Ca^2+^]_i_ transients occurring in cultured MOLT-3 cells were suppressed by mecamylamine, a non-specific nAChR antagonist, or by removal of extracellular Ca^2+^ (Figure 4). On the other hand, scopolamine, an mAChR antagonist; methyllycaconitine (MLA), a specific α7 nAChR antagonist; and nicardipine, a voltage-dependent Ca^2+^ channel blocker; do not affect spontaneous [Ca^2+^]_i_ transients in MOLT-3 cells. This suggests ACh released from MOLT-3 cells activates nAChRs other than the α7 type and induces spontaneous [Ca^2+^]_i_ transients [61]. In addition, PHA-induced TCR activation on MOLT-3 cells enhances ACh synthesis, spontaneous [Ca^2+^]_i_ transients, and expression of the α4 and β2 nAChR subunits. It also suppresses expression of the α9 nAChR subunit. In addition, PHA enhances mRNA expression and synthesis of interleukin (IL)-2 and this effect is suppressed by mecamylamine [61]. Taken together, these findings suggest the involvement of α4β2 nAChRs in the regulation of Ca^2+^ entry and IL-2 synthesis in MOLT-3 cells [61].

In T cells, activation of α7 nAChRs induces metabotropic signaling, resulting in an increase in [Ca^2+^]_i_ without the involvement of ionotropic receptor function [62]. Recent studies also indicate that the α7 and α9/α10 nAChRs expressed in immune cells, including macrophages and T cells, do not function as conventional ligand-gated ion channels; instead, they exert metabotropic functions [62,63,64,65,66,67].

## 4. The Role of ACh in Immune Cells

Our understanding of the function of ACh in the context of immune cell regulation (e.g., mobilization of Ca^2+^ and regulation of cytokine gene expression) is gradually increasing [67,68]. In one recent study, for example, Horkowitz et al. [69] investigated the cholinergic status of lymphocytes in the mouse lung over the course of influenza infection and recovery. They demonstrated the role of ACh in the transition from active immunity to recovery and pulmonary repair following respiratory viral infection. These findings further support the involvement of the lymphocytic cholinergic system in the regulation of immune function.

### 4.1. The Role of α7 nAChRs in the Regulation of Immune Function

The initial observation that α7 nAChR activation prevented the synthesis and release of tumor necrosis factor-α (TNF-α) from activation lipopolysaccharide (LPS)-treated macrophages [68] boosted studies on the roles of α7 nAChRs in the regulation of inflammatory responses. Fujii et al. found that the serum concentration of antigen-specific IgG_1_ was significantly higher in α7 nAChR gene-deficient (α7-KO) mice immunized with ovalbumin (OVA) than in identically treated wild-type (WT) mice [70]. The production of cytokines such as TNF-α, interferon-γ IFN-γ, and IL-6 in cultured spleen cells from α7-KO mice was also significantly greater than in corresponding cells from WT mice. These findings further substantiate the involvement of α7 nAChRs in the regulation of immune function.

The specific function of α7 nAChRs on T cells and antigen-presenting cells (APCs) was further clarified by investigating the effect of GTS-21, a selective α7 nAChR agonist, on the differentiation of CD4^+^ T cells from OVA-specific TCR transgenic DO11.10 mice activated with OVA or with OVA peptide_323−339_ (OVAp) (antigenic epitope) [71]. It was observed that GTS-21 suppressed OVA-induced antigen processing-dependent development of CD4^+^ regulatory T cells (Tregs) and effector helper T (Th1, Th2, and Th17) cells. By contrast, GTS-21 up-regulated OVAp-induced antigen processing-independent development of CD4^+^ Tregs and effector T cells. GTS-21 also suppressed the production of IL-2, IFN-γ, IL-4, IL-17, and IL-6 during OVA-induced activation but enhanced their production during OVAp-induced activation. In addition, during TCR-mediated T cell activation using anti-CD3/CD28 antibodies, GTS-21 promoted the development of all Th lineages, indicating that GTS-21 acts via α7 nAChRs on T cells. Collectively, these findings suggest that (1) α7 nAChRs on APCs suppress CD4^+^ T cell activation by interfering with antigen presentation via inhibition of antigen processing and (2) α7 nAChRs on CD4^+^ T cells promote the development of Tregs and effector T cells. These divergent functions of α7 nAChRs on APCs and T cells are likely associated with the regulation of immune response intensity [67].

CHRNA7, the gene encoding the human α7 nAChR subunit, is located on chromosome 15 and contains 10 exons; exons 1–6 encode the receptor’s extracellular N-terminal region, which contains the ligand-binding domain, while exons 7–10 encode the channel domain region [72]. However, chromosome 15 also contains a human-specific partially duplicated α7 nAChR-like gene with exons 5–10 located 1.6 Mbp 5′ upstream of CHRNA7. Moreover, this human-specific partial duplicate of CHRNA7 is rearranged with a kinase gene (FAM7A) on chromosome 3 to form a hybrid CHRFAM7A [73]. The chimeric CHRFAM7A gene product, termed dupα7, lacks the ligand-binding region but assembles with intact α7 nAChR subunits, resulting in dominant-negative regulation of the channel’s function despite retention of the channel’s structure [74,75,76,77,78]. In addition, Maldifassi MC et al. [79] found that dupα7 sequestration of α7 subunits reduced membrane expression of functional α7-nAChRs. Expression of CHRFAM7A has been detected in human neuronal cells and leukocytes [80] and has been associated with symptoms of schizophrenia and some forms of cognitive deficit (see reviews by Bertrand et al. [81] and Bertrand and Terry [82]. Together, these findings suggest that the failure of α7 nAChR ligands in clinical trials of their efficacy in patients with cognitive deficits, schizophrenia, and other CNS disorders may be ascribed in part to decreased expression of functional α7 nAChRs. Although it has been suggested that α7 nAChR agonists are potentially useful immunomodulatory agents, as described above, their efficacy needs to be confirmed in human immune cells. To evaluate the clinical application of α7 nAChR agonists, however, it will be essential to determine the levels of CHRNA7 and CHRFAM7A expression and to clarify the roles of α7 nAChRs in human T cells and APCs. In contrast to neuronal cells, evidence now suggests that α7 nAChRs on immune cells function as metabotropic receptors rather than as ionotropic receptors [67]. If so, it will be important to investigate the signaling pathways affected by α7 nAChR agonists in the regulation of human immune cell function.

### 4.2. The Role of ACh in Antibody Class Switch

B cells born in the bone marrow differentiate into naïve B cells expressing a particular BCR consisting of membrane-bound IgM and IgD immunoglobulins [83]. Upon activation by their specific foreign antigens and Th cells in peripheral lymphoid organs, naïve B cells proliferate and differentiate into plasma cells or memory cells. Plasma cells produce and release large numbers of different classes of antibodies (IgG, IgA, or IgE) through immunoglobulin class switching.

In addition to nAChRs [84], B cells express all five mAChR subtypes (M_1_–M_5_) [2,3,85]. Oxo-M stimulation of mAChRs on Daudi human B-cell leukemia cells induces an increase in [Ca^2+^]_i_ and Ca^2+^ oscillations mediated by Ca^2+^ release from intracellular Ca^2+^ stores [47,48,61]. Expression of mAChRs on B cells appears to be modulated by activation elicited by immune stimuli [85]. For instance, Pansorbin^®^, which are heat-killed and fixed Staphylococcus aureus with a coat of protein A, trigger Th cell-independent activation of toll-like receptor (TLR) 2 on B cells, leading to mitogenic responses [86]. Additionally, in Daudi cells, 24-h exposure to Pansorbin^®^ results in up-regulated expression of the genes encoding the M_1_-M_4_ mAChRs and the α4nAChR subunit [85]. Similarly, PMA, a PKC activator, plus ionomycin, a calcium ionophore, up-regulate mRNA expression of both M_3_ and M_5_ mAChRs in CCRF-CEM and Daudi cells [87].

Fujii et al. found significantly lower serum concentrations of antigen-specific IgG_1_ in M_1_ and M_5_ mAChR-deficient (M_1_/M_5_-KO) mice than in WT mice, although there was no difference in antigen-specific IgM concentrations between the two genotypes [88]. Furthermore, the synthesis of IL-6, a proinflammatory cytokine, was significantly lower in splenocytes from M_1_/M_5_-KO mice than WT mice. Given that IL-6 promotes the differentiation of B cells into plasma cells [89], these findings suggest the involvement of M_1_ and/or M_5_ mAChRs in the regulation of IL-6-mediated antibody class switching.

Our recent study also showed that when Daudi cells are stimulated with Pansorbin^®^, the resultant mAChR activation enhances class switching from IgM to IgG with a concomitant increase in IL-6 production. On the other hand, the G_q/11_-coupled M_1_, M_3_, M_5_ mAChR-specific inhibitor (4-DAMP) downregulated the production of IL-6 ([85]; Figure 5). Thus, activation of G_q/11_-coupled mAChRs promotes IL-6 production and, consequently, the differentiation of B cells into plasma cells. Importantly, upon binding to Pansorbin^®^, TLR2 signaling via myeloid differentiation primary response protein 88 (MyD88) and MyD88 adaptor-like (MAL)/TIR domain-containing adaptor protein (TIRAP) adaptors activates mitogen-activated protein kinases (MAPKs), including extracellular signal-regulated kinase (ERK) and c-Jun N-terminal kinase (JNK) as well as nuclear factor-κB (NF-κB), which promotes transcription of IL-6 [90]. The increase in [Ca^2+^]_i_ and PKC activation that occurs downstream of G_q/11_-coupled mAChRs may enhance these intracellular signals redundantly, thereby promoting IL-6 production and B cell differentiation into plasma cells.

Furthermore, Skok et al. [91] reported the involvement of α7 nAChRs and other nAChR subunits in the regulation of B cell development and activation. Mouse B cells express several nAChR subtypes, including (α4)_2_(β2)_3_, (α4)_2_(β4)_3_, (α7)_5_, and (α9)_2_(α10)_3_. Moreover, the plasticity of α4β2 and α7 nAChR expression observed upon B cell activation suggests these nAChRs may play a role in the regulation of immune B cell function [92]. Tarasenko et al. [93] found that activation of α7 nAChRs on B cell-derived SP-2/0 cells elicits cation outflux, not Ca^2+^ influx, and an increase in [Ca^2+^]_i_, which suggests the involvement of CRACs in α7 nAChR signaling. Taken together, these findings suggest that by playing various roles in B cells, nAChRs composed mainly of α7 and α4β2 subunits contribute to B cell development, proliferation, and antibody production.

### 4.3. The Roles of ACh in the Regulation of Macrophage Function

Macrophages are immune cells involved in both innate and acquired immunity. When activated by signals from pathogens or living tissues or from environmental stress, macrophages produce physiologically active substances, including the inflammatory cytokine TNF-α, which contributes to the pathogenesis of many inflammatory diseases [94].

Several types of mAChRs and nAChRs are expressed on macrophages [41]. It has been shown, for example, that α7 nAChRs on macrophages are involved in the regulation of TNF-α release and antigen presentation [68,95]. Additionally, it has been confirmed that human macrophage-like U937 cells express M_1_, M_3_, and M_5_ mAChRs as well as the α4, α7, α9, and β4 nAChR subunits. Moreover, expression of all five mAChR subtypes and the α2, α4, α5, α6, α7, α10, and β2 nAChR subunits have been detected in mouse peritoneal macrophages [41,96]. Although it is known that α7 nAChRs on macrophages contribute to the regulation of cholinergic anti-inflammatory pathways by mediating the suppression of inflammatory cytokines [3,5,97], the contributions made by other nAChR subunits to the regulation of macrophage function have not been reported, nor has the contribution of mAChRs.

## 5. Effects of Two Peptide Ligands on the Cholinergic Activity in T Cells

Cholinergic activity in T cells is modulated by several factors, including the activation of PKA and PKC [28] and the general level of immunological stimulation, such as that mediated by CD3 or CD11a [9,98]. Here, we describe the effects of two peptide ligands exhibiting cholinergic activity in T cells.

### 5.1. Effect of Secreted Lymphocyte Antigen-6/Urokinase-Type Plasminogen Activator Receptor-Related Peptide (SLURP)-1

SLURP-1 was initially identified as a secretory protein in the Ly6 superfamily, which has structural similarity to neurotoxins from snakes and frogs, including, for example, muscarinic toxin α and Xenoxin-1. Most of the family members are glycosylphosphatidylinositol (GPI)-anchored cell surface proteins [99]. However, SLURP-1 is a secreted protein, and although it has three-dimensional structural similarity to the potent α7 nAChR antagonist α-bungarotoxin, it acts as a positive allosteric ligand potentiating the action of ACh on α7 nAChRs [100]. Interestingly, mutations in the gene encoding SLURP-1 have been detected in patients with Mal de Meleda (MdM), a rare autosomal recessive skin disorder characterized by transgressive palmoplantar keratoderma [101]. In that context, SLURP-1 was shown to stimulate the pro-apoptotic activity and differentiation in human keratinocytes via putative (but not yet defined) allosteric sites on α7 nAChRs [102]. Moreover, SLURP1-deficient mice show severe palmoplantar keratoderma characterized by increased proliferation of keratinocytes and water barrier defects [103]. These findings suggest that SLURP-1 regulates epidermal homeostasis through α7 nAChRs.

In addition, one study using T cells from MdM patients with a genetic SLURP-1 mutation showed that SLURP-1 plays an essential role in T cell activation [104]. In fact, SLURP-1 mRNA has been detected in most immune organs, including the thymus and spleen [105], as well as peripheral blood MNLs, DCs, and macrophages [41,105,106]. This suggests that SLURP-1 is involved in the regulation of immune function.

Detection of SLURP-1 immunoreactivity in CD205^+^ DCs surrounded by CD4^+^ T cells in the interfollicular zone of human tonsils suggests the possibility that SLURP-1 released from CD205^+^ DCs participates in the enhancement of signals transduced via α7 nAChRs during antigen presentation [107]. PHA, which activates T cells via TCR signaling, attenuates proliferation of MOLT-3 human leukemic T cells and increases their ACh content, while MLA, a selective α7 nAChR antagonist, abolishes all effects elicited by PHA [107]. Like PHA, recombinant SLURP-1 also attenuated MOLT-3 cell proliferation and increased their ACh content, and those effects, too, were abolished by MLA. This suggests that SLURP-1 activates cholinergic transmission by potentiating ACh synthesis and, in turn, stimulation of α7 nAChRs, thereby facilitating the functional development of T cells. This finding supports the notion that SLURP-1 acts as a key positive allosteric or agonistic modulator of immune responses [107].

Recent findings also revealed that SLURP-1 inhibits the growth of multiple cancer cell lines [108,109]. This is in line with the observation that SLURP-1 stimulates pro-apoptotic activity in human keratinocytes and attenuates the proliferation of MOLT-3 human leukemic T cells [107]. SLURP-1 activates the intracellular domain of α7 nAChRs, leading to the recruitment and phosphorylation of Janus kinase 2 (JAK2) and the subsequent activation of signal transducer and activator of transcription proteins 3 (STAT3). [110,111]. Although the precise pathway leading to the induction of SLURP-1 is not fully understood, these findings suggest SLURP-1 should be considered for the development of new treatment approaches for cancer.

### 5.2. Effect of Hippocampal Cholinergic Neurostimulating Peptide (HCNP)

Originally purified from juvenile rat hippocampus, HCNP is an undecapeptide secreted after cleavage of the N-terminal domain of the 21 kDa HCNP precursor protein (HCNPpp) [112,113]. In rat medial septal nucleus cholinergic neurons, HCNP up-regulates ChAT gene expression and thus promotes ACh synthesis [114,115]. Notably, expression levels of HCNP and its precursor protein are decreased in the brains of dementia patients (including Alzheimer’s disease-derived and ischemic vascular dementia) [116]. As ACh is important for the regulation of glutamatergic neuronal activity and for the development of hippocampal neurons, HCNP appears to be essential for the coordination and maintenance of neuronal activity through ACh synthesis and may, therefore, contribute to brain-related pathogenic processes.

HCNPpp is widely distributed at both the protein and mRNA levels in the central nervous system, especially in the hippocampus, cerebellum, and hypothalamus [113]. However, it is also present at various levels in peripheral tissues, including the testis, liver, kidneys, and spleen [117], from which HCNPpp appears to be secreted. In fact, the concentration of HCNPpp in bovine serum is estimated to be approximately 35 nM [118]. HCNP is reportedly stored in the secretory granules of chromaffin cells within the medulla of the adrenal glands together with catecholamines, which are released in response to nAChR activation [118,119]. It appears that HCNP serves as a ligand to activate M_2_ mAChRs in the heart, exerting a negative inotropic effect under basal conditions and counteracting the adrenergic positive inotropism [119].

Importantly, HCNPpp is also found in CD4^+^ and CD8^+^ T cells in the spleen and in macrophages [120,121]. Long-term (5 days) exposure to HCNP decreases expression of ChAT in the MOLT-3 human T cell line, thereby suppressing ACh synthesis [120]. Similarly, in human macrophage-like U937 cells, HCNP suppresses LPS-induced ChAT expression [121], which may be attributable to the attenuation of ERK expression and its phosphorylation. HCNP is, therefore, thought to negatively regulate ChAT expression in immune cells. This difference from neuronal cells is probably associated with there being several types of ChAT transcripts (R-type, N1, N2, and M-type), which originate from three distinct promoter regions and reflect alternative splicing of 5′-noncoding exons [122]. The most abundant transcript in the brain and spinal cord is the M-type, followed by the R-type and N-types. On the other hand, immune cells mainly express the N2-type, though T cells reportedly express R-type, N1, N2, and M-type ChAT [123].

HCNP has been associated with suppression of LPS-induced inducible NO synthase (iNOS) and COX-2 expression in U937 cells [121]. In addition, HCNPpp, which is also known as phosphatidylethanolamine binding protein (PEBP) and Raf-1 kinase inhibitory protein (RKIP), can as its names imply, modulate several cellular signaling pathways [113]. HCNPpp interacts with Raf-1 and mitogen-activated protein extracellular kinase (MEK) to inhibit MAPK/ERK and NF-κB pathways. It is possible that HCNP retains some of the functionality of HCNPpp and attenuates the LPS-induced transcription of these inflammatory enzymes. Alternatively, the HCNP-induced decreased in ChAT gene expression may reduce ACh synthesis in U937 cells, indirectly suppressing inflammatory enzymes. In addition, activation of G_q/11_-coupled mAChRs reportedly enhances the release of cytokines after exposure to LPS. In summary, although the precise mechanism of HCNP is still unknown, it has been suggested that HCNP attenuates cholinergic signals through downregulation of ChAT expression in immune cells. However, because HCNP also suppresses expression of inflammatory mediators, it should be considered an immunosuppressant agent.

## 6. Conclusions

The findings presented in this review provide clear evidence that lymphocyte functions are regulated by a cholinergic system via both mAChRs and nAChRs. Ca^2+^ mobilization in T cells, B cells, and macrophages is a key signaling pathway leading to the regulation of immunity. In T cells, for instance, activation of both mAChRs and nAChRs, together with the observed increase in [Ca^2+^]_i_ and upregulation of c-fos and IL-2 mRNA expression, strongly support the hypothesis that ACh synthesized in and released from T cells acts as an autocrine and/or paracrine regulator of immune function. In B cells, moreover, mAChR activation enhances immunoglobulin class switching from IgM to IgG. Finally, in macrophages, the anti-inflammatory effects of ACh mediated via α7 nAChRs are well established. Interestingly, two peptides, SLURP-1 and HCNP, are known to contribute to the regulation of the lymphocyte cholinergic system (positively or negatively). Additional new findings related to non-neuronal cholinergic systems in immune cells will provide insight not only for a better understanding of immune regulatory mechanisms, but also for the development of novel immunomodulatory drugs.

## Figures and Tables

**Figure 1 ijms-22-06818-f001:**
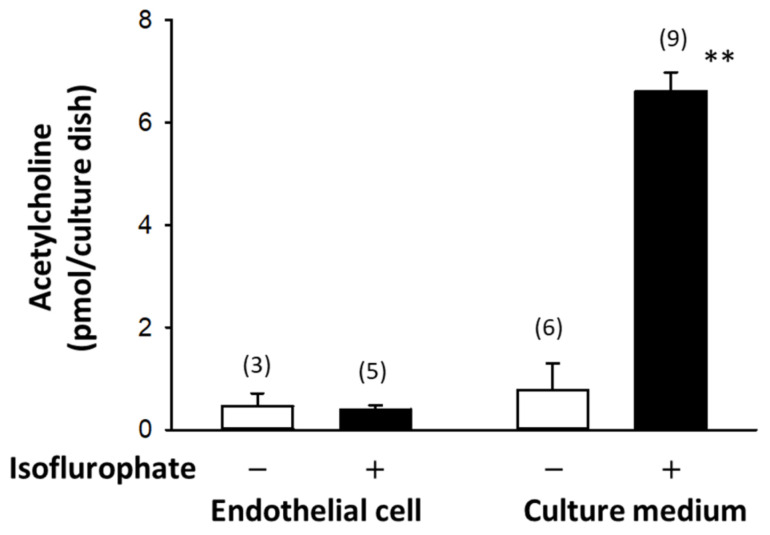
Synthesis and release of ACh in bovine arterial endothelial cells cultured for 24 h with or without isoflurophate. Isoflurophate strongly protected ACh from degradation. ** *p* < 0.01 vs. the value obtained in the absence of isoflurophate. Bars indicate means standard error of means. The numbers of samples are shown in parentheses. Rearranged from [20].

**Figure 2 ijms-22-06818-f002:**
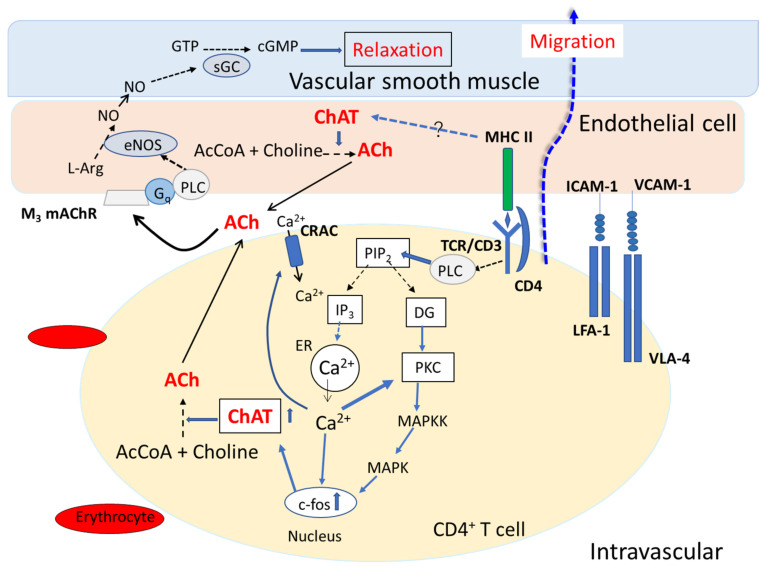
Schematic drawing illustrating the interaction between a vascular endothelial cell (VEC) and a CD4^+^ T cell, leading to cholinergic activation and transendothelial migration of the T cell. Interaction between vascular endothelial cells (VECs) and CD4^+^ T cells is mediated by specific interactions between major histocompatibility complex II (MHC II) and the T cell receptor/CD3 complex (TCR/CD3), between intercellular cell adhesion molecule-1 (ICAM-1) and lymphocyte function-associated antigen-1 (LFA-1), and between vascular cell adhesion molecule-1 (VCAM-1) and very late antigen-4 (VLA-4). Stimulation of TCR/CD3 signaling pathways in CD4^+^ T cells activates phospholipase C (PLC), leading to conversion of phosphatidyl 4,5-bisphospahe to inositol 1,4,5-trisphosphate (IP_3_) and diacylglycerol (DG). IP_3_ induces Ca^2+^ release from the endoplasmic reticulum (ER). The released Ca^2+^ induces influx of extracellular Ca^2+^ through the calcium release-activated channel (CRAC). DG with Ca^2+^ activates protein kinase C (PKC). PKC activates mitogen-activated protein kinase (MAPK) kinase (MAPKK), which in turn activates MAPK. MAPK promotes c-fos mRNA expression in the nucleus, leading to up-regulation of choline acetyltransferase (ChAT) mRNA expression (see reviews by [23,28]). Little is known about the regulatory mechanisms governing ACh synthesis and release in VECs. However, given that MAPK activation and c-Fos protein levels are closely connected in MHC-II signaling [29], one could speculate that TCR-induced transmembrane signaling elicited by MHC II enhances ChAT mRNA expression in VECs. ACh synthesized from choline and acetyl coenzyme A (AcCoA) in a reaction catalyzed by ChAT in both VECs and CD4^+^ T cells is released and acts on M_3_ mAChRs expressed on the surface of VECs. The M_3_ mAChR activation leads to nitric oxide synthase (NOS)-catalyzed production of NO from L-arginine (L-Arg). NO activates soluble guanylate cyclase (sGC), which catalyzes the production of cGMP and relaxes the vascular smooth muscle. Relaxation of vascular smooth muscle facilitates the extravasation of CD4^+^ T cells.

**Figure 3 ijms-22-06818-f003:**
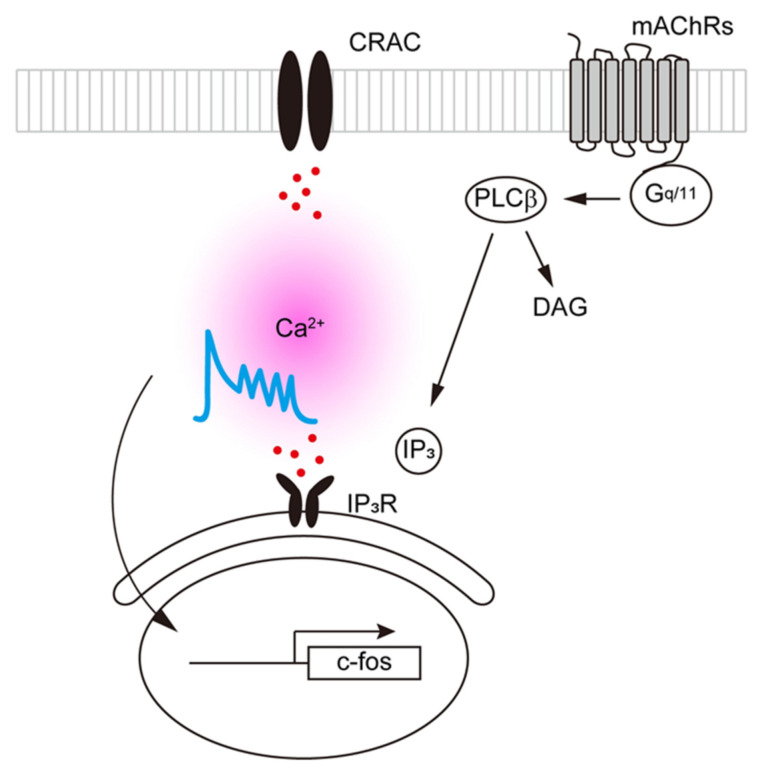
G_q/11_-coupled mAChRs evoke [Ca^2+^]_i_ increases in T cells. G_q/11_-coupled M_3_ and/or M_5_ mAChR activation results in IP_3_-mediated Ca^2+^ release from the endoplasmic reticulum followed by extracellular Ca^2+^ influx through CRACs, which mediate repetitive [Ca^2+^]_i_ oscillations in T cells. The increase in [Ca^2+^]_i_ is required to enhance c-fos gene expression.

**Figure 4 ijms-22-06818-f004:**
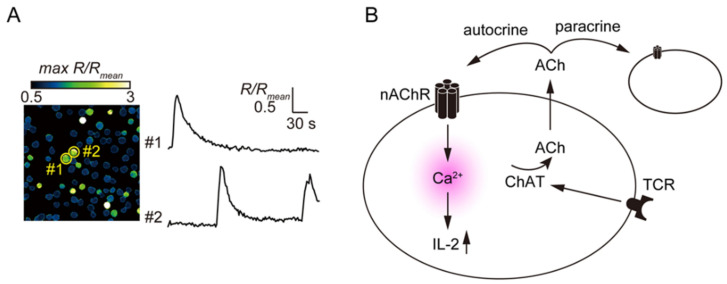
Autocrine ACh acts via nAChRs to elicit [Ca^2+^]_i_ transients in MOLT-3 cells. (**A**) Spontaneous increases in [Ca^2+^]_i_ within MOLT-3 cells during a 5-min observation period. Representative traces showing the spontaneous [Ca^2+^]_i_ responses. Reproduced with permission from [61]. (**B**) ACh synthesized by ChAT and then released from T cells activates nAChRs on the surfaces of themselves or neighboring cells, which results in transient [Ca^2+^]_i_ increases and IL-2 synthesis. TCR activation enhances the autocrine action of ACh by mediating increased ChAT expression.

**Figure 5 ijms-22-06818-f005:**
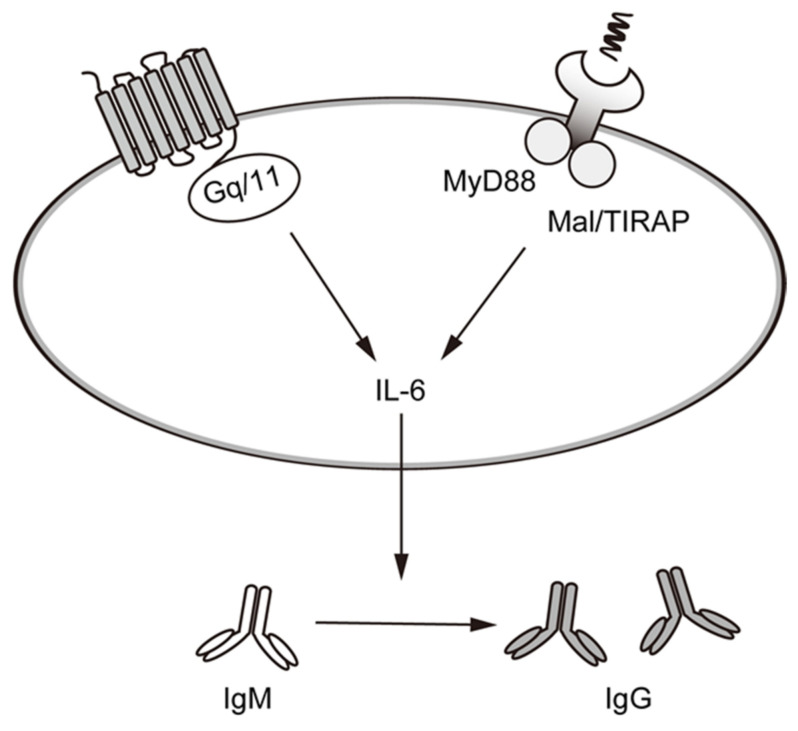
Activation of mAChRs and nAChRs enhances IL-6 release and immunoglobulin class switching from IgM to IgG in Daudi cells. Cooperative activation of mAChRs and TLR2 promotes the production of the IL-6 responsible for B cell maturation. In Daudi cells, this leads to class switching from IgM to IgG. Signals from nAChRs are important for activating the MyD88 and Mal/TIRAP pathways, which in turn increases the production and release of IL-6.

## Data Availability

Not applicable.

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
