# Peer review of "Regulation of Immune Functions by Non-Neuronal Acetylcholine (ACh) via Muscarinic and Nicotinic ACh Receptors"

_ijms, 2021, doi:10.3390/ijms22136818_

Round 1
Reviewer 1 Report
Title: Delete “non-neuronal” as it suggests a different type of ACh exists in neuronal and non-neuronal systems.
Line 40: Upper-case for each first letter of “international”, “symposium” and “non-neuronal”
Line 92: Delete “non-neuronal”
Line 98: Delete “non-neuronal”
Line 100: Add “(see section 3.1)” after “M3 mAChRs” as it would be helpful.
Line 119-122: Move “Isoflurophate … Rearranged from [20}” to the end of line 118 sentence.
Line 167-175: What is the significance of the underlined words?
Line 170: Delete “inositol-1,4,5-triphosphate” as it is already abbreviated in previous section
Line 248-251: Move “Gq/11…expression” to the end of line 247 sentence.
Line 255: Replace “differ” with “in”
Line 262: Add “α subunits or” after “only”
Line 288-293: Move “A. Spontaneous…expression” to the end of line 287 sentence.
Line 414-417: Move “Cooperative…release of IL-6” to the end of line 413 sentence.
Author Response
Ms#: IJMS-1275099
Review #1
- Title: Delete “non-neuronal” as it suggests a different type of ACh exists in neuronal and non-neuronal systems.
Line 92: Delete “non-neuronal”
Line 98: Delete “non-neuronal”
Response: The term “non-neuronal acetylcholine (ACh)” is now well accepted in life sciences to distinguish it from neuronally-derived ACh. Because ACh represents one of the best characterized neurotransmitters, some readers of this journal may not be aware of neuroanatomical evidence that indicates little possibility of interaction of neuronally-derived ACh with ACh receptors on immune cells. Therefore, we have employed the term “non-neuronal acetylcholine”.
- Line 40: Upper-case for each first letter of “international”, “symposium” and “non-neuronal”
Response: In accordance with the reviewer’s suggestion, we have corrected the sentence.
- Line 100: Add “(see section 3.1)” after “M3mAChRs” as it would be helpful.
Response: Thanks. We have added “(section 3.1)” after “M3 mAChRs”
- Line 119-122: Move “Isoflurophate … Rearranged from [20}” to the end of line 118 sentence.
Response: Thanks. We have moved “Isoflurophate….”. In addition, we have rewritten the second sentence of the Figure 1 legend as follows: “Isoflurophate strongly protected ACh from degradation”
- Line 167-175: What is the significance of the underlined words?
Response: Thanks. The new Figure 2 (previous Figure 1) have been rewritten completely. Therefore, the explanations on the new Figure 2 have been rewritten and underlined in the revised manuscript.
- Line 170: Delete “inositol-1,4,5-triphosphate” as it is already abbreviated in previous section
Response: Thanks. We have replaced “inositol-1,4,5-triphosphate (IP3)” with “IP3”.
- Line 248-251: Move “Gq/11…expression” to the end of line 247 sentence.
Response: Thanks. We have moved “Gq/11…expression” to the end of line 247 sentence.
- Line 255: Replace “differ” with “in”
Response: Thanks. We have replaced “differ” with “among”.
- Line 262: Add “α subunits or” after “only”
Response: Thanks. We have added “subunits” after “only a”.
- Line 288-293: Move “A. Spontaneous…expression” to the end of line 287 sentence.
Response: Thanks. We have moved “A. Spontaneous…expression” to the end of line 287 sentence.
- Line 414-417: Move “Cooperative…release of IL-6” to the end of line 413 sentence.
Response: Thanks. We have moved “Cooperative…release of IL-6” to the end of line 413 sentence.
Reviewer 2 Report
The resubmitted version of the manuscript is fine and can be endorsed for publication
Author Response
Thank you.
This manuscript is a resubmission of an earlier submission. The following is a list of the peer review reports and author responses from that submission.
Round 1
Reviewer 1 Report
The authors provide wide evidence on the sources and roles of non-neuronal acetylcholine in regulating mainly T lymphocyte functions. Interesting evidence is provided on the endogenous regulators of cholinergic activity, like SLURP-1 and HCNP; the latter being quite new for immune cells. The authors are known experts in this field and, in fact, they review the results of their own studies with respect to the data of others supporting or forming a background of their studies. The 105 cites references include 42 papers of the authors’ team. This makes the review quite targeted but also results in some gaps not covered by the authors’ studies. In particular, relatively minor attention is paid to the role of cholinergic regulation in innate immunity and in B lymphocytes. Meanwhile, there is plenty of data on macrophages, neutrophils and natural killer cells, as well as on B lymphocytes.
The authors’s statement that “no anatomical evidence is available to support the formation of synapses between cholinergic nerve endings and immune cells” (lines 123-124) is not fully correct because cholinergic innervation of the bone marrow and thymus, but not the spleen, has been demonstrated (Bellinger et al., 1993; Singh and Fatani, 1988; Artico et al., 2002). Moreover, it was shown that both cutting and stimulating the cholinergic nerves affected the hemopoiesis in the bone marrow and thymus (Chernigovskiy et al., 1967; Singh and Fatani, 1988). Therefore, in addition to non-neuronal acetylcholine, immune cells can be affected by acetylcholine released from the nerve endings, at least, in the course of development.
The statement that “α7 nAChRs expressed in B cells do not affect B cell differentiation or immunoglobulin class switching” (line 379) is also not fully correct because it was shown that both α4β2 and α7 nAChRs support the survival of B cell precursors and increase the size of B-lymphocyte population in the bone marrow (Skok et al., 2006). a7 nAChRs expressed in mature B lymphocytes are coupled to CD40, which mediates B lymphocyte activation and immunoglobulin class switch. Moreover, a7 nAChRs of both T and B lymphocytes were found recruited to immune synapse formed between these cells, suggesting that acetylcholine is an additional mediator to modulate activation of interacting T and B lymphocytes (Koval et al., 2011).
Although the nAChR signaling in immune cells is mainly metabotropic (as absolutely correctly mentioned by the authors of the manuscript), either acetylcholine or α7-specific agonist PNU282987 stimulated the ion currents in B lymphocyte-derived hybridoma cells. However, Ca2+ influx was observed only within minutes, was mediated by CRACs and could be stimulated by either PNU282987 or MLA indicating that α7 nAChRs influence CRACs in ion-independent way (Tarasenko et al., 2020).
Minor comments
Abstract, line 26: IS not only neurotransmitter;
Line 96 – too many comas;
Lines 125-126: “ACh diffuses into immune cells”. ACh rather affects the cells from outside by influencing nAChRs and mAChRs.
Line 156: "mAChRs and AChRs" (“n” is missing).
Lines 338-340: “Plasma cells undergo class switching to produce and release large amounts of different classes of antibodies (IgG, IgA, or IgE) from membrane-bound IgM” : should be re-phrased.
Reviewer 2 Report
I agreed to review this paper because the cholinergic control of inflammation has come to be appreciated as a very important topic for investigation. The two search terms "cholinergic" and "inflammation" draw 5215 hits on PubMed, more than half of which have been in the last ten years. A search for the "cholinergic anti-inflammatory pathway" produces 5725 hits. Unfortunately, for as much as this is an important topic, the review by Mashimo et al. is pretty much off the mark. It is overloaded with disconnected observations and speculations based on reports of correlations converted into assumptions of causality. There is no synthesis or presentation of a "big picture" or models that would connect the many dots that are all over the place. A basic figure of leucocyte cell lineages and the factors that regulate leucocyte differentiation would have been very useful. A figure illustrating how precisely the authors theorize ACh plays into these pathways would have also been an invaluable, albeit absent, contribution. Where does the ACh signal come from, the vagus, or is it only autocrine? What stimulates the ACh production? What are the translational implications? Instead of a synthesis of current data in the field, the paper delivers a string of citations largely based on their own work (over 37 self-citations), primarily from relatively low-profile journals.
Aside from Figure 1, which is about endothelial cells not immune cells, the figures in the paper are blatant republications of previous findings, and not very interesting findings at that. Figure 4 is from Doshisha Women's College of Liberal Arts annual reports of studies, which is not even from a legitimate journal! The other three figures, all from Mashimo et al. papers, have received a total of three citations from other groups since they were published.
In my opinion, this "review" offers nothing of value to an important field of research.
Other comments:
The authors show a very limited understanding nAChR function and diversity and hence this paper is insufficient to edify anyone on this topic.
Line 100: It is hard to believe that a radioimmunoassay for acetylcholine in the rat brain would be very useful, especially since its used requires isofluorophate, a dangerous organophosphate poison.
Line 304: I think it is very unlikely that the existence of the dup-alpha7 (CHRFAM7A) is the reason for clinical failures.
Line 320: "With this respect, it is worth investigating the effects of alpha7 nAChR agonists on the regulation of human immune cell function."
With probably over 400 publications on this topic, largely ignored by this review, I would say that is rather an understatement.
Lines 470-471: "We hypothesize that SLURP-1 activates the catalytic intracellular domain of alpha7 nAChRs, leading to the recruitment and phosphorylation of JAK2 and the subsequent activation of STAT3". Just what catalytic intracellular domain are they talking about?
The tangents on SLURP-1 and HCNP are disconnected from the rest of the data and so speculative that they really add nothing.
Reviewer 3 Report
Line 18: Replace Ach with “ACh”
Line 25: Delete parenthesis after “(HCNP)”
Line 42: Delete extra space after “[4]”
Line 64: Replace dependent with “independent”
Line 66: "Torpedo" should be "Torpedo"
Line 96: Delete comma after “released”
Figure 1: M3 AChR should be labelled as “M3 mAChR”
Line 104: Add “by” before “ChAT”
Line 105: First mention of M3 mAChR. Perhaps a brief mention of the mAChR subunits in section 1.3?
Line 107: Replace “is leading to” with “in turn”
Table 1: Legend mentions “**P <0.01” but not shown in Table 1.
Line 113: “Arranged from [20]” should read “Rearranged from [20]”
Line 156: Replace “AChR” with “nAChR”
Line 171 and 173: “fura-2” should be “Fura-2”
Line 175: Replace “a mAChR” with “an mAChR”
Line 180: Superscript “2+”
Line 201: As CCRF-CEM cells express both M3 and M5 mAChRs, “…upon M3 or M5…” should read “…upon M3 and/or M5…”
Line 204: As c-fos is used throughout the review, “c-Fos” should be “c-fos”
Line 211: Replace “were” with “have been”
Line 220: The statement “Their activation by nicotine…” is not accurate for all nAChRs as nicotine is an antagonist at α9-containing subtypes (e.g. homomeric α9 and heteromeric α9α10). Delete “nicotine or other”.
Line 223: Add “-mentioned” after “above”
Line 223-224: Replace “the expression of mRNAs of muscle-types nAChRs subunits…” with “the mRNA expression of muscle type nAChR subunits”
Line 230: Replace “a mAChR” with “an mAChR”
Line 243: “fura-2” should be “Fura-2”
Line 301 and 302: “α7 nAChR agonists” should read “α7 nAChR-selective agonists”
Line 316: “agents acting on” should read “agents selective for”
Line 321: “α7 nAChR agonists” should read “α7 nAChR-selective agonists”
Line 355: Replace “MyD88” with “myeloid differentiation primary response protein 88 (MyD88)”
Line 355: Replace “MAL/TIRAP” with “MyD88 adaptor-like (MAL)/TIR domain-containing adaptor protein (TIRAP)”
Line 356: Replace “JNK” with “c-Jun N-terminal kinase (JNK)”
Line 363: “Daudi cell .A. Representative” should read “Daudi cells. A. Representative”
Line 420-425: Misplaced figure legend (?)
Line 436: Replace “GPI” with “Glycosylphosphatidylinositol (GPI)”
Line 472: “JAK2” should read “Janus kinase 2 (JAK2)”
Line 472: “STAT3” should read “signal transducer and activator of transcription proteins 3 (STAT3)”